# Effects of Incorporating Different Proportions of Humic Acid into Phosphate Fertilizers on Phosphorus Migration and Transformation in Soil

Jianyuan Jing [1,†], Shuiqin Zhang [2,†], Liang Yuan [2], Yanting Li [2], Yingqiang Zhang [2], Xinxin Ye [1,*], Ligan Zhang [1], Qizhong Xiong [1], Yingying Wang [1] and Bingqiang Zhao [2,*]

1    Anhui Province Key Laboratory of Farmland Ecological Conservation and Pollution Prevention, School of Resources and Environment, Anhui Agricultural University, Hefei 230036, China; jyjing@ahau.edu.cn (J.J.); zhligan@ahau.edu.cn (L.Z.); qzxiong@ahau.edu.cn (Q.X.); 18226102404@163.com (Y.W.)
2    Key Laboratory of Plant Nutrition and Fertilizer, Ministry of Agriculture and Rural Affairs, Institute of Agricultural Resources and Regional Planning, Chinese Academy of Agricultural Sciences, Beijing 100081, China; shuiqin08@163.com (S.Z.); yuanliang@caas.cn (L.Y.); liyanting@caas.cn (Y.L.); zhangyqysu123@163.com (Y.Z.)
*    Correspondence: yexx@ahau.edu.cn (X.Y.); zhaobingqiang@caas.cn (B.Z.); Tel.: +86-551-65786311 (X.Y.); +86-01-82108568 (B.Z.)
†    These authors contributed equally to this work.

**Abstract:** Incorporating humic acid (HA) into phosphate fertilizers to produce HA-enhanced phosphate fertilizers (HAPs) can improve the migration and availability of fertilizer-derived phosphorus (P) in soil. However, the optimal proportion of HA remains inconsistent. Here, we investigated the effects of HAPs with different HA proportions (0.1–10% $w/w$) on water-soluble P fixation rate, P migration, P transformation, and soil microorganisms, and analyzed the main P forms in HAP using Fourier transform infrared spectroscopy and nuclear magnetic resonance spectroscopy. The results showed that incorporating 0.1% HA had no impact on P migration and transformation, whereas incorporating 0.5–10% HA increased the migration distance and cumulative migration of fertilizer-derived P by 0–5 mm and 17.1–30.3%, respectively, compared with conventional phosphate fertilizer (CP). Meanwhile, HAPs with 0.5–10% HA significantly reduced the water-soluble P fixation rate by 18.3–25.6%, and significantly increased the soil average available P (AP) content in 0–40 mm soil layer around the P application site by 6.2–12.9% relative to CP, partly due to the phosphate monoesters in HAPs. Clustering analysis revealed that 0.5% HA had similar effects relative to higher HA proportions (1% and 5%), and the inhibition of HAP with 0.5% HA on bacteria and fungi was also greater than that of CP due to the high concentration of soil P. Overall, 0.5% was determined to be the optimal amount of HA for HAP production, which provided a theoretical basis for the development of high-efficiency phosphate fertilizer.

**Keywords:** humic acid-enhanced phosphate fertilizer; incorporating proportion; humic acid; phosphorus migration; phosphorus availability

## 1. Introduction

P is an indispensable element for plant growth and development [1]. Despite the fact that the application of phosphate fertilizers meets the crop's demand for P and increases agricultural productivity [2], the availability of P in soil is often limited due to most of the added fertilizer P experiencing rapid fixation [3,4]. Meanwhile, the high affinity of P for the soil solid phase also results in the poor migration of fertilizer P in soil [5]. As a result, a substantial amount of fertilizer P is not absorbed and utilized by plants [6,7], reducing P use efficiency and agricultural productivity [8,9]. To address this issue, various agents have been developed, such as coated controlled-release phosphate fertilizers [10], phosphate fertilizers containing humic acid (HA)–metal–phosphate complexes [11], biological phosphate

fertilizers [12], HA- or alginic acid-enhanced phosphate fertilizers [13], and other modified phosphate fertilizers [14]. Among these, the incorporation of HA into phosphate fertilizers to produce HAPs has led to significant improvements in crop yield and P use efficiency relative to CP [15–17].

The success of HAPs in agricultural production is largely attributed to the positive effects of HA in reducing P fixation, increasing P availability, and promoting P migration in soil [18–21]. There are three probable mechanisms by which HA reduces P fixation and improves P availability. First, HA can bind to $Ca^{2+}$ or $Mg^{2+}$ in calcareous soils and $Fe^{3+}$ or $Al^{3+}$ in acidic soils, thereby preventing the formation of insoluble precipitates [22,23]. Second, HA can adhere to the surface of insoluble phosphate and effectively inhibit the precipitation of phosphate minerals [24]. Third, HA can form stable and water-soluble phosphate–metal–HA complexes, which aid in reducing P fixation and improving P availability [22]. Meanwhile, previous research has also demonstrated that HA can promote P migration in soil [25,26], which is beneficial to the absorption of fertilizer P by roots [6].

The use of HAs to regulate P migration and transformation for improving P availability is supported by the findings of the studies cited above. However, the optimal proportion of HAs in HAPs remains unclear, with proposed levels varying widely. Shafi et al. [27] observed that applying single superphosphate with 45 kg HA $ha^{-1}$ significantly improved soil water-soluble P contents compared with a single superphosphate application alone at P increments from 45 to 112.5 kg $P_2O_5$ $ha^{-1}$. Chen et al. [10] reported that the combination of controlled-release P fertilizers and 45 kg HA $ha^{-1}$ increased soil AP content by 38.6% compared with controlled-release P fertilizer alone at 75 kg $P_2O_5$ $ha^{-1}$. Du et al. [25] also found that applying monocalcium phosphate fertilizer with 254.8 kg HA $ha^{-1}$ (at a rate equivalent to 60.9 kg $P_2O_5$ $ha^{-1}$) could increase the distance of P movement by 5 mm. However, Zhao et al. [15] suggested that trace amounts of HAs (<1% or around 2 kg HA $ha^{-1}$, based on 75 kg $P_2O_5$ $ha^{-1}$) in phosphate fertilizer were sufficient to produce strong positive effects on P migration and transformation. Notably, the structural composition and physicochemical properties of HA in HAPs are significantly altered during HAP production [28], leading to the improved biological activity of HA in promoting plant growth [29]. Therefore, the amount of HA incorporated into phosphate fertilizer warrants further investigation, as the high performance of HAPs is also attributable to HA-mediated stimulation of plant growth, and the amount of HA is a crucial factor affecting plant growth [30,31].

In the present study, the effects of different proportions of HA (0.1%, 0.5%, 1%, 5%, and 10% *w/w*) in HAP formulations on water-soluble P fixation rates, fertilizer-derived P migration, soil P availability, and soil microbial community structure were investigated. In addition, the forms of P in HAPs were analyzed using Fourier transform infrared (FTIR) spectroscopy and solid-state $^{31}P$ nuclear magnetic resonance ($^{31}$P-NMR) spectroscopy to reveal the mechanism of HA regulating P fixation. The objective of this study was to determine the ideal amount of HA to be incorporated into phosphate fertilizers, and the findings could serve as a reference for developing optimal HAPs.

## 2. Materials and Methods

### 2.1. Preparation of HA, CP, and HAPs

HA was extracted from weathered coal collected from Huolinhe, Tongliao, Inner Mongolia Autonomous Region, Northeast China (45°23′ N, 119°15′ E) using the methods described by Zhang et al. [32]. Additionally, in China, the above weathered coal was the main raw material used to produce HA for HAP. Based on the elemental analyzer (Vario Micro Cube, Elementar Analysensysteme GmbH, Frankfurt, Germany), the contents of carbon (C), hydrogen (H), oxygen (O), nitrogen (N), and sulfur (S) in HA were 59.5%, 2.7%, 31.1%, 2.5%, and 0.8%, respectively. Meanwhile, the relative contents of HA fractions with molecular weights below 10 kDa, 10–100 kDa, and above 100 kDa were 62.8%, 15.4%, and 21.7%, respectively, as determined by gel permeation chromatography (GPC, Shimazu LC-20A, Kyoto, Japan). Additionally, the relative proportions of C-containing functional groups of HA were measured using solid-state $^{13}$C nuclear magnetic resonance spectrometry (Bruker

AVANCE III HD 400 MHz, Switzerland), and the aromatic C (aromatics and aromatic C-O) and the carboxyl C were the most dominant C types, accounting for 66.9% and 20.6%, respectively. The detailed data of the above parameters were presented in Table 1.

**Table 1.** Elemental composition, the percentage of molecular weight, and the relative proportions of C-containing functional groups of HA.

| Element Composition (%) | | | | | The Percentage of Molecular Weight (%) | | |
|---|---|---|---|---|---|---|---|
| C | H | O | N | S | <10 kDa | 10–100 kDa | >100 kDa |
| 59.5 | 2.7 | 31.1 | 2.5 | 0.8 | 62.8 | 15.4 | 21.7 |
| **Relative proportions of C-containing functional groups (%)** | | | | | | | |
| Alkyl C | O-alkyl | Aromatics | Aromatic C-O | | Carboxyl C | | Ketones/Aldehydes |
| 2.0 | 6.5 | 58.8 | 8.1 | | 20.6 | | 4.2 |

Data are from Jing et al. [28].

To prepare CP, the industrial production process for dipotassium phosphate was simulated in the laboratory. In brief, 53.3 g of potassium hydroxide was added to a beaker containing 46.7 g of 85% (*v/v*) phosphoric acid with continuous stirring, and the CP was obtained after pulverizing and grinding the reaction product through a 0.85 mm sieve.

HAP was prepared following the same procedure used for CP, except that a certain amount of HA was added to the phosphoric acid before adding potassium hydroxide. In this study, HAPs with 0.1%, 0.5%, 1.0%, 5.0%, and 10.0% HA were prepared, and the corresponding products were named HAP0.1, HAP0.5, HAP1, HAP5, and HAP10, respectively.

The content of HA and water-soluble P in CP and HAPs were determined using the methods described in "mono ammonium phosphate or diammonium phosphate containing humic acid" (HG/T 5514-2019) published by the Ministry of Industry and Information Technology of China. The contents of total P (TP) and potassium (K) in the phosphate fertilizers were determined using the vanadium molybdate yellow colorimetric method [33] and flame photometry [34] following wet digestion with $H_2SO_4$-$H_2O_2$ [35]; Table 2 shows the corresponding data.

**Table 2.** Humic acid (HA), P, and K content of phosphate fertilizers.

| Phosphate Fertilizer | HA Content (%) | | Total P ($P_2O_5\%$) | Water-Soluble P ($P_2O_5\%$) | The Ratio of Water-Soluble P (%) | Total K ($K_2O\%$) |
|---|---|---|---|---|---|---|
| | Theoretical Contents | Actual Contents | | | | |
| CP | 0.00 | 0.00 | 35.80 | 34.00 | 94.97 | 54.38 |
| HAP0.1 | 0.10 | 0.09 | 35.71 | 33.98 | 95.16 | 54.27 |
| HAP0.5 | 0.50 | 0.50 | 34.57 | 33.63 | 97.28 | 53.68 |
| HAP1 | 1.00 | 0.99 | 34.47 | 33.59 | 97.45 | 52.77 |
| HAP5 | 5.00 | 4.85 | 32.36 | 31.42 | 97.10 | 49.64 |
| HAP10 | 10.00 | 9.73 | 30.83 | 29.78 | 96.59 | 48.16 |

Data are the average of three measurements.

## 2.2. Structural and Compositional Analysis of CP and HAP10

The structural and compositional characteristics of HAPs were analyzed using FTIR spectroscopy and $^{31}$P-NMR spectroscopy. Given that the amount of HA in HAPs was relatively low, it was difficult to distinguish between HAP and CP. Therefore, we only selected HAP10 and CP for spectral analysis. FTIR spectra from 4000 to 400 cm$^{-1}$ were determined using an FTIR spectrometer (Nicolet iS10, Thermo Nicolet Corporation, Waltham, MA, USA) with a resolution of 4 cm$^{-1}$ and 32 scans per sample. For $^{31}$P-NMR analysis, spectra were collected using an NMR spectrometer (JNM-ECZ600R, JEOL, Tokyo, Japan), and chemical shifts of phosphates were identified using $K_2HPO_4$ and $K_3PO_4$ as references. The

spectra were recorded using a 3.2 mm probe at a spin rate of 10 kHz, a resonance frequency of 242.95 MHz, and 1024 scans per sample.

### 2.3. Water-Soluble P Fixation Rate of CP and HAPs

To compare the effects of incorporating different proportions of HA into phosphate fertilizer on P fixation, the water-soluble P fixation rates of CP, HAP0.1, HAP0.5, HAP1, HAP5, and HAP10 were determined using the $CaCl_2$ precipitation method [36]. The water-soluble P fixation rate (%) was calculated as follows:

$$\text{Water} - \text{soluble P fixation rate} = \frac{(P_1 - P_2)}{P_1} \times 100 \tag{1}$$

where $P_1$ is the initial P content of phosphate fertilizer ($P_2O_5$%), and $P_2$ is the residual P content of phosphate fertilizer after reacting with $CaCl_2$ ($P_2O_5$%).

### 2.4. Migration of P Derived from CP and HAPs

#### 2.4.1. Soil Incubation

To compare the migration of fertilizer P in soil treated with HAPs with different HA content, soil incubation experiments were conducted using soil samples collected over 9 years in Yucheng County, Dezhou City, Shandong Province, China ($36°50'$ N, $116°34'$ E; altitude 21.2 m) from the 0–20 cm soil layer in the field without fertilizer input. The soil was fluvo-aquic soil classified as light loam. The collected soil samples were air-dried and crushed to a particle size of <2 mm. The basic physical and chemical properties of the soil were as follows: organic matter, 4.1 g kg$^{-1}$; pH 8.02 [soil to distilled water ($w/v$) at a 1.0:2.5 ratio]; soil TP, 0.7 mg g$^{-1}$; AP, 10.2 mg kg$^{-1}$; $NH_4^+$-N, 1.0 mg kg$^{-1}$; $NO_3^-$-N, 1.4 mg kg$^{-1}$; and available K, 79.7 mg kg$^{-1}$.

The soil incubation experiments were conducted using a specially designed cuboid container (Figure 1A) made of polyurethane with dimensions of 7 cm length, 10 cm height, and 7 cm width and a movable baffle called "Side A" for easy soil sampling. Seven treatments, including CP, HAP0.1, HAP0.5, HAP1, HAP5, HAP10, and a no-fertilizer control treatment (NCT), were applied, with three replicates for each treatment. Except for NCT, the P application rate was 3 g $P_2O_5$ kg$^{-1}$ of soil dry weight [19]. The soil moisture content was adjusted to 50% of the field capacity and precultured in a climate chamber in the dark at 25 °C for 10 days. The soil moisture content was then adjusted to 60% of field capacity, and each incubation container, containing 688 g of soil with a soil bulk density of 1.23 g cm$^{-3}$, was subpackaged. During the addition process, the soil was compacted in layers, and the final height was 9.5 cm. The phosphate fertilizers were accurately weighed and evenly spread on the soil surface, and the container was covered with pierced plastic wrap to prevent excess water evaporation and ensure ventilation. The containers were then moved into a climate chamber and incubated under dark conditions at 25 °C. Distilled water was added using a weighing method to maintain the soil moisture level at around 60% of field capacity.

#### 2.4.2. Soil Sampling and Laboratory Analyses

After 30 days of incubation, the container was rotated 90° so that "Side A" was at the top, after which "Side A" was removed (Figure 1A). The entire soil block was carefully moved horizontally out of the container, and the soil was sliced every 5 mm from top to bottom using a special steel wire (Figure 1B,C). The soil layer at a vertical distance of 8.5–9.5 cm from the fertilization layer was not cut, resulting in the collection of 18 soil samples from each container. The soil samples were air-dried, ground to pass through a nylon sieve, and sealed in plastic bags for the determination of TP (≤0.15 mm) using concentrated $H_2SO_4$–$HClO_4$ extraction [37] and AP (≤0.85 mm) using $NaHCO_3$ solution extraction [38] via the molybdate–ascorbic acid method.

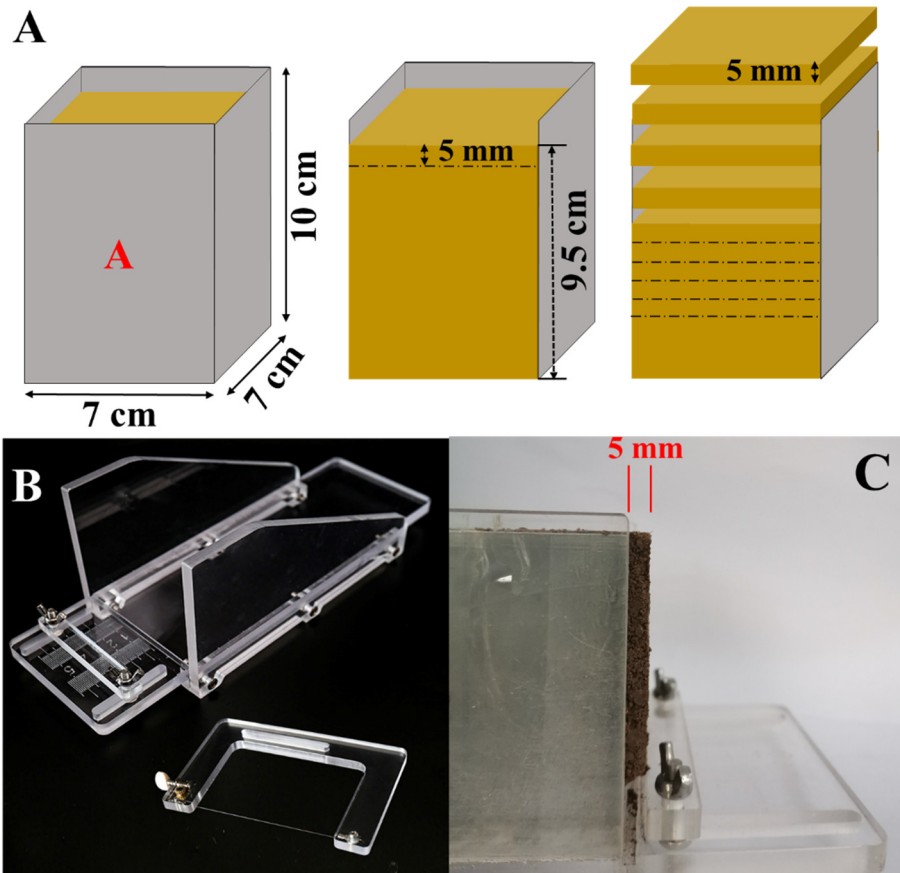

**Figure 1.** Schematic diagram of soil culture container (**A**), soil cutting tools (**B**), and soil cutting method (**C**).

### 2.4.3. Calculations

The amount of fertilizer P in the $i$th soil layer (MAFP; g P) ($i$ = 1 to 18, from top to bottom), the cumulative percentage of fertilizer P migration (CPFPM), and soil P activation coefficient (PAC) was calculated as follows:

$$\text{MAFP} = (\text{P}_{Total} - 0.72) \times 1.23 \times 24.5 \times 10^{-3} \tag{2}$$

$$\text{CPFPM} = \frac{\sum_{i=1}^{n} \text{MAFP}_i}{\text{FP}} \tag{3}$$

$$\text{PAC} = \frac{\text{P}_{Avail}}{\text{P}_{Total} \times 10^3} \times 100 \tag{4}$$

where $\text{P}_{Total}$ is the soil TP content in the $i$th soil layer (g P kg$^{-1}$), 0.72 is the average content of soil TP in NCT (n = 54; g P kg$^{-1}$), FP is the amount of fertilizer P applied (g P), $\text{P}_{Avail}$ is soil AP content in the $i$th soil layer (mg P kg$^{-1}$), 1.23 is soil bulk density (g cm$^{-3}$), 24.5 is the soil volume of each soil layer (cm$^3$), $10^{-3}$ converts g to kg, and n is the ordinal number of soil layers counting from the fertilization layer ($1 \leq n \leq 18$).

### 2.4.4. Phospholipid Fatty Acid (PLFA) Analysis

For NCT, CP, and HAP0.5 treatments, the soil microbial community composition and microbial biomass in the 0–40 mm soil layer from the P application site were determined by PLFA analysis, according to the procedure described by Ai et al. [39]. Five grams of soil were randomly selected from each of the eight blocks cut from the 0–40 mm soil layer, and then mixed them to obtain the tested soil. The concentrations of PLFAs were expressed in units of nmol g$^{-1}$. Total microbial biomass was estimated using the total

concentration of PLFAs. PLFAs were divided into various taxonomic groups based on previously published PLFA biomarker data [40]. Specifically, we used 15:0iso as Gram-positive bacteria (Gm+) biomarkers; 16:1$\omega$7c as Gram-negative bacteria (Gm−) biomarkers; and the sum of Gm+ and Gm− biomarkers together with 15:0anteiso, 17:0iso, 17:0anteiso, 17:0cyclo w7c, 18:1w7c, 19:0cyclo w7c, and 18:1w5c as a measure of total bacterial biomass. The unsaturated PLFAs 16:1 w5c, 18:1$\omega$9c, and 18:2$\omega$6c were used as fungal biomarkers. The fatty acids 10Me-16:0 were used as markers for actinomycetes. Additionally, the ratio of 18:1$\omega$9c and 18:2$\omega$6c to total bacterial PLFAs was the ratio of fungi to bacteria (F:B).

*2.5. Mobility of HA in HAPs*

The effect of HA on fertilizer-derived P migration may be related to its mobility in the soil. However, the additional amount of HA in HAP was too low to determine the mobility of HA by measuring its content in different soil layers. To assess the mobility of HA in soil, white quartz sand with a particle size of 0.15–0.50 mm was used as a substitute for deep soil, and the variation in quartz sand color was used as an indicator. Specifically, 2 g of soil was mixed with a specific amount of CP or HAPs and then poured into a 20-mL glass tube (1.5 cm in diameter and 14.5 cm in height) containing 22 g of quartz sand. The application rate of CP or HAPs was 3 g $P_2O_5$ kg$^{-1}$ of the total weight of soil and quartz sand. Subsequently, 5 mL of distilled water was slowly added, and the tube was covered with pierced plastic wrap to prevent excess water evaporation and ensure ventilation. All glass tubes were incubated in the dark at 25 °C in a climatic chamber, and the color of quartz sand was observed after 24 h.

*2.6. Statistical Analysis*

The variation among the different treatments was analyzed using SAS 9.1 (SAS Institute Inc., Cary, NC, USA). To evaluate the difference between treatments, analysis of variance was used with the least significant difference (LSD) test ($\alpha$ = 0.05). Based on the results of soil average AP content and CPFPM, hierarchical cluster analysis of all P fertilizers was conducted using the nearest neighbor method in Origin 2022 (Origin Lab Corporation, Northampton, MA, USA). Principal component analysis (PCA) was used to compare and analyze the structure of the soil microbe community, which was indicated by the composition of PLFAs.

## 3. Results

*3.1. Orthophosphate and Phosphate Ester Exists in HAPs*

We analyzed the FTIR spectra of CP and HAP10 (Figure 2A) to determine the presence of both phosphate and organic functional groups in HAPs. The results indicated that HAP10 had a similar FTIR spectrum to CP, with characteristic peaks including (a) H-bonded OH or adsorbed-water stretching vibration at 3500–3300 cm$^{-1}$, (b) PO-H stretching vibration at 3300–3200 cm$^{-1}$, (c) $PO_4^{3-}$ antisymmetric or P = O stretching vibration at 1077 cm$^{-1}$, (d) P-OH stretching vibration at 987 cm$^{-1}$, and (e) $HPO_4^{2-}$ variable-angle vibration or $PO_4^{3-}$ asymmetric variable-angle vibration at 859, 660, and 539 cm$^{-1}$ [41,42]. To identify differences between HAP10 and CP in the FTIR spectra, we compared their second derivative spectra (Figure 2B). Our findings showed that the vibration intensity at 981, 864, and 536 cm$^{-1}$ was higher for CP than for HAP10, whereas the vibration intensity at 1083 cm$^{-1}$ was higher for HAP10 than for CP (Figure 2B). These results suggest that the content of orthophosphate in HAP10 was lower than that in CP. However, phosphate ester C-O-P stretching vibration might have existed in HAP10 at 1083 cm$^{-1}$ [36].

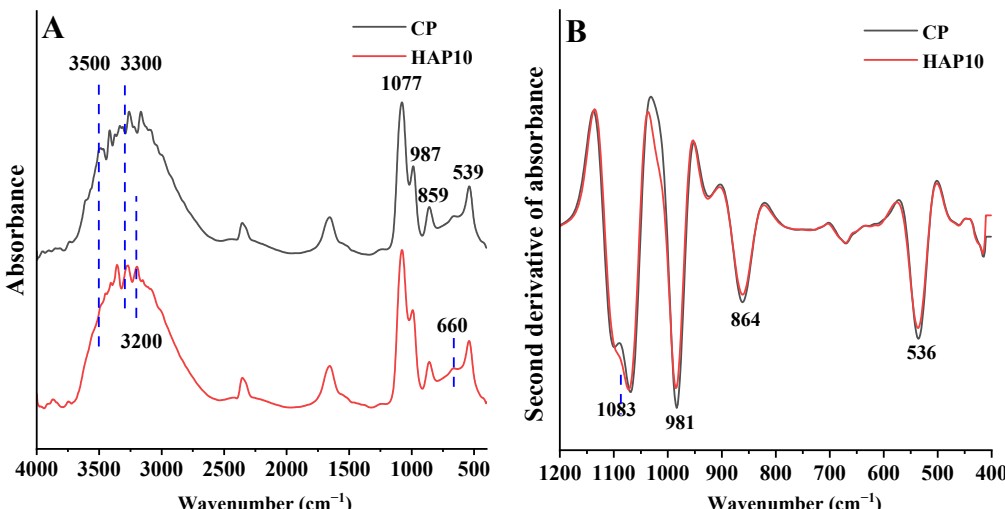

**Figure 2.** FTIR spectra (**A**) and second derivative spectra (**B**) of CP and HAP10 fertilizer.

The presence of phosphate esters in HAPs was confirmed by generating solid-state $^{31}$P NMR spectra for CP and HAP10 (Figure 3). The spectra of reference phosphates indicated that the signal peaks of $K_2HPO_4$ and $K_3PO_4$ were at 5.20 and 12.34 ppm, respectively. In the spectrum of CP, we observed a shift in the peak at 5.22 ppm, indicating the formation of $K_2HPO_4$, the reaction product of phosphoric acid and potassium hydroxide. However, the spectrum of HAP10 showed three single peaks at 11.40, 5.99, and 4.04 ppm. This result confirmed that in addition to the orthophosphates of $K_2HPO_4$ (5.99 ppm) and $K_3PO_4$ (11.40 ppm), other forms of P existed in HAP10. Combined with the FTIR second derivative spectra (Figure 2B), this suggests that phosphate esters (4.04 ppm) were present in HAP10.

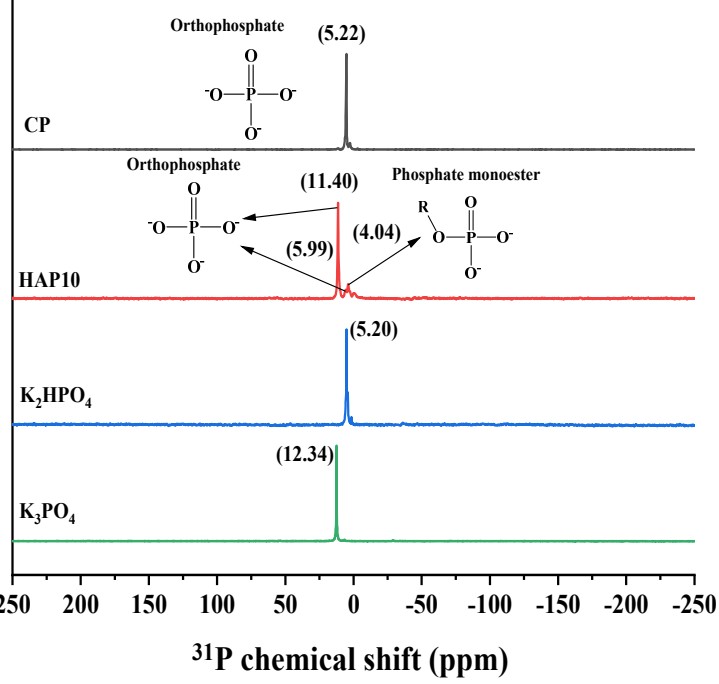

**Figure 3.** Solid−state $^{31}$P−NMR spectra of CP and HAP10 fertilizer.

### 3.2. HA Combined with Phosphate Fertilizer Reduces the Rate of Water-Soluble P Fixation

We compared the water-soluble P fixation rate of various HAPs with that of CP alone (Figure 4), and found that the addition of HA to phosphate fertilizer at any tested proportion (HAP0.1–HAP10) resulted in a 6.3–25.6% lower water-soluble P fixation rate

relative to CP alone ($p < 0.05$). In addition, the water-soluble fixation rate decreased with the amount of HA added. These findings demonstrate that HA in HAP can effectively prevent fertilizer-derived P from being immobilized (Figure 4).

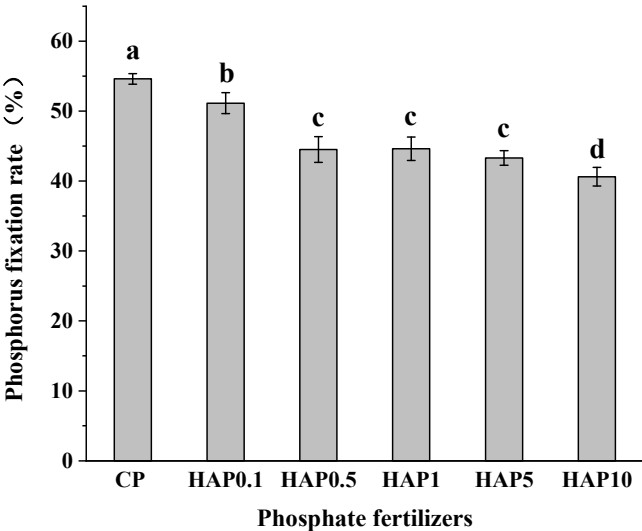

**Figure 4.** Water-soluble P fixation rate of phosphate fertilizer with different addition amount of HA. The additional amounts for HAP0.1, HAP0.5, HAP1, HAP5, and HAP10 fertilizers were 0.1%, 0.5%, 1%, 5%, and 10% *w/w* HA, respectively. Treatments with no letter in common were significantly different at $p < 0.05$, as indicated by the LSD test.

### 3.3. HAP Application Promotes the Migration of Fertilizer-Derived P

To quantitatively assess the impact of HA proportions in HAP on fertilizer-derived P migration, the distribution of TP content was measured by slicing the soil into layers (Figure 1C). Under CP or HAP treatments, the TP content in the soil was initially high near the P application layer but sharply decreased beyond a distance of 40 mm from the fertilizer site. The TP content of all P application treatments reached the same level as the control treatment when the distance was >55 mm (Figure 5A). Comparing the TP content at 45–50 mm and 50–55 mm soil layers from the fertilization layer (Figure 5B,C), all P application treatments resulted in 19.5–100.5% higher TP content compared with that in the control at 45–50 mm ($p < 0.05$; Figure 5B), with TP content significantly higher in all HAP treatments (except for HAP0.1) relative to the CP treatment ($p < 0.05$; Figure 5B). However, only the HAP0.5, HAP1, HAP5, and HAP10 treatments resulted in significantly increased TP content (11.0%, 7.4%, 10.1%, and 37.3%, respectively) compared with NCT when the distance was 50–55 mm ($p < 0.05$; Figure 5C), and there was no significant difference between the CP or HAP0.1 treatment and the control ($p > 0.05$). Overall, these findings indicate that the addition of HA to phosphate fertilizers could increase the migration distance of fertilizer-derived P, the migration distance of fertilizer P for HAP0.5, HAP1, HAP5, and HAP10 was 0–5 mm higher than that for CP.

We next calculated the effects of HA on the amount of fertilizer-derived P migration. Given that the migration of fertilizer P occurred in the top 55 mm of soil from the fertilizer site (Figure 5A), we compared the CPFPM variation of CP and HAPs within the 0–55 mm soil depth (Figure 6). CPFPM increased in a linear-plus-plateau manner with increasing soil depth, and the plateau value of CPFPM and its corresponding soil depth differed among P application treatments (Figure 6). The soil depth corresponding to the peak of CPFPM was defined as the "inflection point distance", which was 40.1, 40.0, 43.9, 44.0, 44.5, and 44.7 mm for CP, HAP0.1, HAP0.5, HAP1, HAP5, and HAP10, respectively, with CPFPM peaks of 74.3%, 76.3%, 87.0%, 90.1%, 92.4%, and 96.8%, respectively (Figure 6). Compared with CP, the amount of fertilizer P in HAPs that migrated increased by 17.1–30.3% (Figure 6), excluding HAP0.1, which only increased by 2.7% (Figure 6B). These results further demon-

strate that the addition of HA to phosphate fertilizers increases the migration distance and amount of fertilizer P, with the effect being proportional to the amount of HA added.

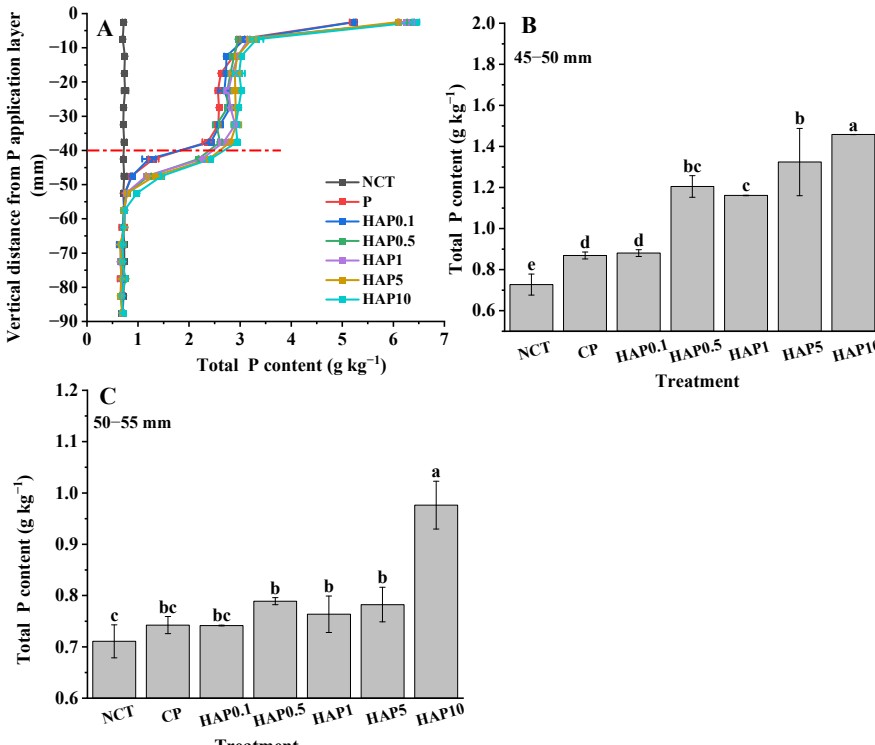

**Figure 5.** Distribution of soil total P content in the soil layer 0–90 mm from the P application layer (**A**), and comparison of soil total P content in the soil layers 45–50 mm (**B**), and 50–55 mm (**C**) from the P application layer. Treatments with no letter in common were significantly different at *p* < 0.05, as indicated by the LSD test.

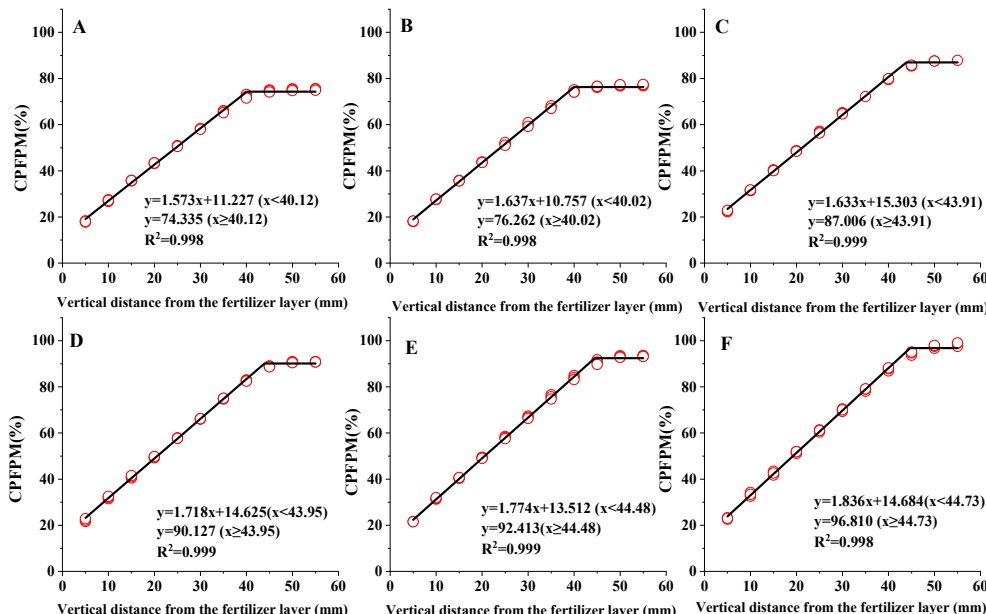

**Figure 6.** Variation in the cumulative percentage of fertilizer P migration (CPFPM) with increasing vertical distance from fertilization layer within 0–55 mm. (**A**–**F**) were the variation curves of CP, HAP0.1, HAP0.5, HAP1, HAP5, and HAP10 fertilizers, respectively.

### 3.4. HAP Application Enhances Soil P Availability

Based on the effects of HAPs on fertilizer-derived P migration, we investigated their effects on soil P availability. Consistent with soil TP content distribution, soil AP content decreased sharply beyond a distance of 40 mm from the fertilizer site and reached the same level as the control when the distance was >55 mm (Figure 7A). We further evaluated soil AP levels in the 0–40 mm soil layer and found that CP and HAP application significantly increased soil average AP content relative to NCT. Moreover, compared with CP application, HAP treatment (except for HAP0.1) resulted in a 6.2–12.9% higher average AP content ($p < 0.05$; Figure 7B). A comparison of soil AP content in the 50–55 mm soil layer revealed that HAP treatments (except for HAP0.1) resulted in significantly higher AP content than that resulting from CP treatment (Figure 7C). However, only HAP10 treatment significantly increased TP content relative to CP treatment in the 50–55 mm soil layer (Figure 5C). Furthermore, in the 45–50 mm and 50–55 mm soil layers from the fertilization site, the soil PAC values of HAP0.5, HAP1, HAP5, and HAP10 were significantly higher than those of CP ($p < 0.05$; Table S1). These results suggest that the addition of 0.5–10% HA to P fertilizer enhances P availability and delays the process of P transformation from effective to ineffective.

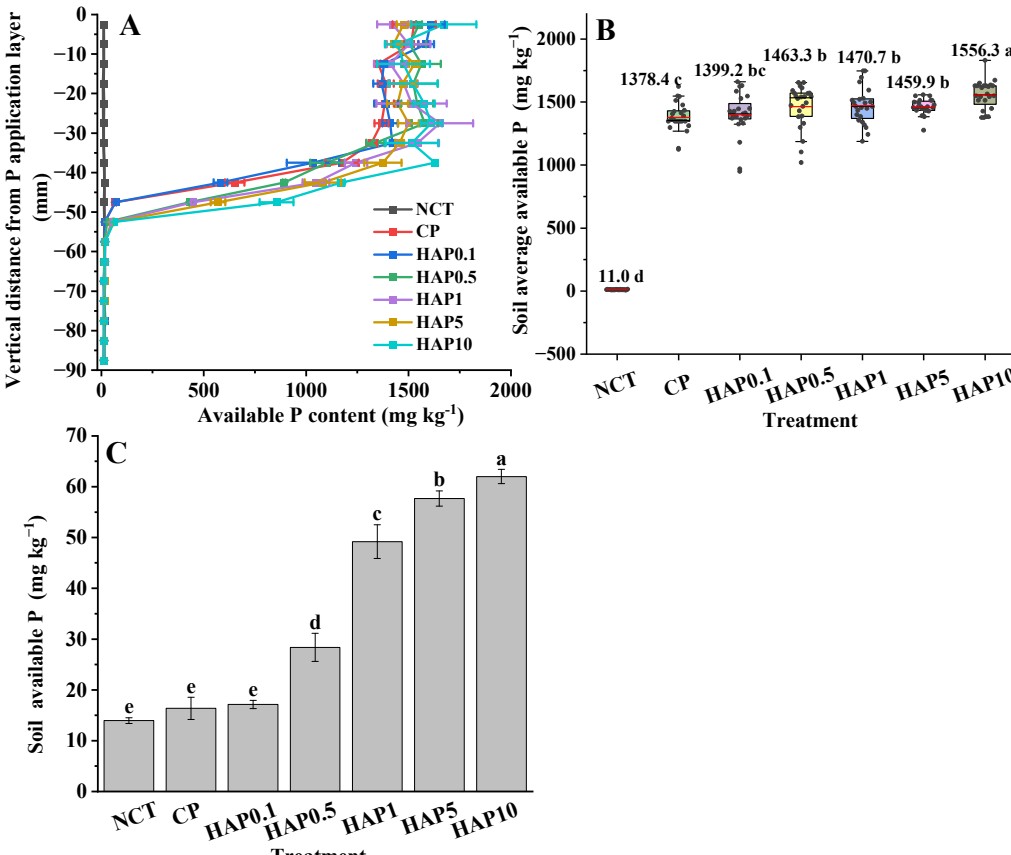

**Figure 7.** Distribution of soil available P content in the soil layer 0–90 mm from the P application layer (**A**), comparison of soil average available P content throughout the soil layers in 0–40 mm (**B**), and comparison of soil available P content in the soil layers 50–55 mm from the P application layer (**C**). (**B**), the black and red lines, lower and upper edges of the boxes, and bars and dots outside the boxes represent median and mean values, 25th and 75th, 5th and 95th, and <5th and >95th percentiles of all data, respectively. Values represent the average (n = 24). Treatments with no letter in common were significantly different at $p < 0.05$, as indicated by the LSD test.

### 3.5. HA Movement Distance in HAPs Increases with the Amount of HA Added

We conducted a simulation to evaluate the migration ability of HA derived from HAP in soil, using quartz sand as a soil substitute. Our findings revealed that HAP0.5, HAP1, HAP5, and HAP10 changed the color of quartz sand to different degrees, resulting in effects at depths of 7.5, 8.4, 9.0, and 10.6 cm, respectively, whereas the color of quartz sand treated with CP and HAP0.1 remained essentially unchanged (Figure 8). These results confirm that HA in HAP has mobility in soil, and when the amount of HA added reaches 0.5% $w/w$, the movement distance is considerable. Thus, the promotion of P migrants and enhancement of P availability via the addition of HA to P fertilizer may be related to the movement of HA in the soil.

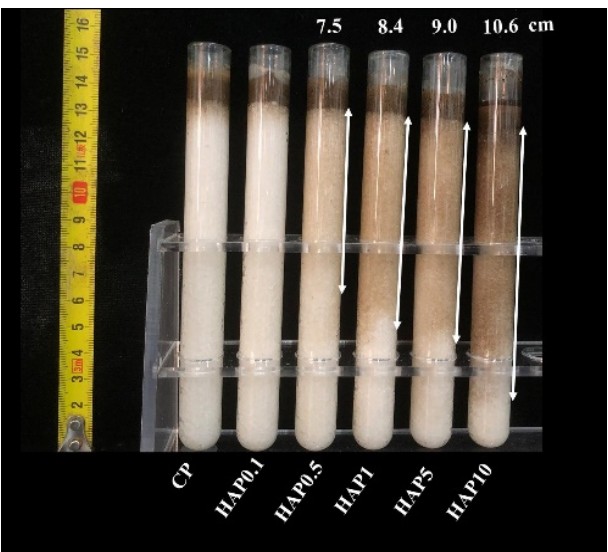

**Figure 8.** Mobility of HA derived from HAPs in quartz sand simulated soil. The white number at the top represents the distance that the quartz sand color changes.

## 4. Discussion

### 4.1. Phosphate Monoester Is Formed during HAP Production

In this study, we investigated the composition of HAP and CP based on FTIR spectra (Figure 2B) and solid-state $^{31}$P NMR spectra (Figure 3). We found that HAP contained phosphate ester in addition to orthophosphate, whereas CP contained only orthophosphate, and that phosphate ester was formed during HAP production. Zhang et al. [36] used citric acid, the model component of HA, to prepare citric acid-enhanced phosphate fertilizer and found a high-temperature esterification reaction between the –OH of citric acid and P-OH bond of phosphoric acid, as well as the simultaneous production of phosphate monoester. Therefore, the high temperature generated by the acid–base neutralization reaction promoted the esterification of phosphoric acid and HA in HAP preparation, wherein the oxygen atom in HA bonded with the P atom to form phosphate monoester (P–O–C). Li et al. [43] also found that phosphate monoester could be obtained via the esterification of phosphoric acid and hydroxyl groups. The presence of organophosphorus in HAP is conducive to reducing the fixation of fertilizer-derived P and enhancing the bioavailability of soil P [44–46].

In the present study, owing to the reaction between HA and phosphoric acid, the amount of phosphoric acid was reduced, resulting in the formation of $K_2HPO_4$ and excess potassium hydroxide, which reacted and formed $K_3PO_4$. As a result, the HAP10 fertilizer contained three P forms, and the $K_2HPO_4$ single peak in $^{31}$P-NMR spectra was weak (Figure 3).

*4.2. Incorporating HA into Phosphate Fertilizer Promotes P Migration and Enhances P Availability*

Numerous studies have demonstrated the effectiveness of HA-incorporated phosphate fertilizer on promoting P migration. For instance, An et al. [6] found that the AP content of the soil with HA-coated phosphate fertilizer exceeded that with CP at different distances from the application site. Similarly, Du et al. [25] reported that HA applied with monocalcium phosphate could promote fertilizer P migration. These findings align with our results, indicating that HA incorporated into phosphate fertilizer enhances P migration (Figures 5A and 6). The mobility and anti-P fixation function of HA is responsible for its ability to promote fertilizer P migration [6,18,22]. In the present study, we used white silica as a soil substitute to investigate the mobility of HA in HAP fertilizer and found that HA has mobility in the medium, with a considerable HA moving distance observed when the amount added reached 0.5% $w/w$ (Figure 8).

The promotion of fertilizer P migration by HA was also associated with the fact that HA could reduce the fixation of fertilizer P. In this study, the soil average AP content in 0–40 mm soil layers was significantly higher with HAPs (except HAP0.1) than with CP (Figure 7B). Additionally, HA delayed the process of P transformation from effective to ineffective (Figure 7C and Table S1), and the water-soluble P fixation rates of HAPs were significantly lower than that of CP (Figure 4). These findings indicate that HA combined with P fertilizer reduces P fixation and increases soil available P content [17,47]. This enhancement appeared to be supported by the interaction between HA and metal cations of soil, which effectively avoided the binding of $PO_4^{3-}$ and metal ion and improved the availability of soil P [19,24]. The formation of the HA–metal–phosphate bridge complex is also essential for reducing P fixation and improving P availability and migration in the soil [22,48,49]. Additionally, organically bound P fertilizers, such as HAPs containing organophosphorus (Figures 2B and 3), are less prone to soil fixation and retain relatively high mobility compared with inorganic P fertilizers [46]. This factor could also contribute to the greater P migration distances and soil available P levels achieved with HAPs relative to CP.

We also found that the water-soluble P fixation rate of HAP0.1 was significantly lower than that of CP (Figure 4), but there was no significant difference between HAP0.1 and CP in terms of soil average AP content (Figure 7B) and fertilizer P migration (Figure 6A,B). These results indicate that the level of HA added to HAPs was too low to prevent fertilizer P fixation effectively in soil. Moreover, the regulating efficacy of HAs in HAPs on fertilizer-derived P migration and transformation was related to the amount of HA added.

*4.3. Optimal Use of 0.5% HA in HAPs Promotes P Migration and Transformation*

To determine the ideal proportion of HA in HAPs for promoting fertilizer-derived P migration and transformation in soil, hierarchical cluster analysis was conducted. The results showed that HAP0.1 was clustered with CP, indicating that 0.1% HA was insufficient to promote fertilizer-derived P migration in soil, which was consistent with the weaker effects of HAP0.1 in improving P migration and availability (Figures 5A and 7B). Conversely, HAP10 showed the highest CPFPM peak (Figure 6F) and soil average available P content (Figure 7B) within the 0–40 mm soil layer, indicating its superiority in terms of promoting P migration and enhancing P availability. Therefore, HAP10 was separated into a distinct class after cluster analysis (Figure 9). The remaining HAPs were clustered together, indicating that differences in the effects of HAs at 0.5%, 1%, and 5% on fertilizer P migration and transformation were limited. Based on the cost and production process of HAPs, 0.5% HA might be the optimal amount for incorporation into P fertilizer.

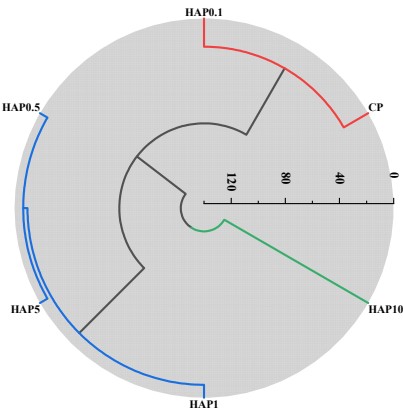

**Figure 9.** Hierarchical cluster analysis of CP and HAP fertilizers based on the peak of CPFPM, the "Inflection point distance", and the soil average available P content in 0–40 mm soil depth.

Furthermore, by comparing the soil community structure in the 0–40 mm soil layer around the P application site, it was found that both CP and HAP0.5 significantly reduced the total biomass of PLFA and inhibited the growth of bacteria (mainly Gm+) and fungi (Table S2), which were mainly related to the higher soil P concentration. It is generally accepted that high soil P levels could inhibit the growth of microorganisms [50,51]. In this study, we also found that the total PLFA, bacteria, fungi biomass, and F:B ratio were inversely correlated with TP or AP (Table S3). Therefore, the degree of inhibition of microbial growth could be used to assess the level of soil P concentration. In this study, the PCA results showed that HAP0.5 inhibited bacteria (15:0iso, 15:0anteiso, 17:0iso, 17:0anteiso, 17:0cyclo w7c, 18:1w7c, and 19:0cyclo w7c) or fungi (18:1ω9c and 18:2ω6c) to a greater extent than CP (Figure 10), further indicating that the addition of 0.5% HA in phosphate fertilizer was sufficient to significantly improve the availability of soil P. In addition, previous studies have shown that HAP0.5 exerted significant beneficial effects on crop yield and P use efficiency [15,17,47], and our results also support these findings. However, our results only showed that phosphate fertilizer had a negative impact on the soil microbial community structure in tiny areas surrounding it, and would not affect the growth of soil microorganisms in the whole soil due to the limited P migration distance.

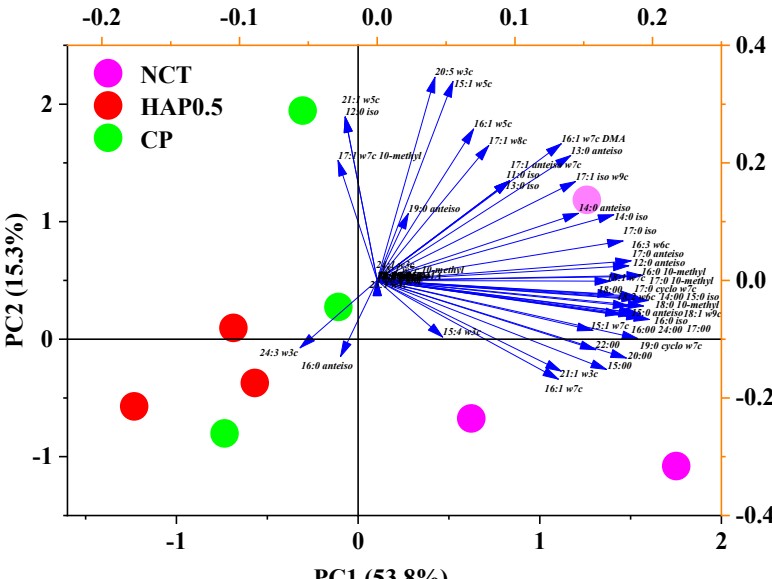

**Figure 10.** Principal component analysis (PCA) of soil microbial community within 0−40 mm soil layer around the P application site under NCT, CP, and HAP0.5 treatments.

## 5. Conclusions

Incorporation of HAs into phosphate fertilizers at proportions of 0.5%, 1%, 5%, and 10% ($w/w$) resulted in significant beneficial effects on the amount and distance of P migration, as well as on soil average AP levels. The ability of HAPs to improve P migration and transformation in soil can be partly attributed to the formation of phosphate monoesters during their production. Among the various HAP formulations, HAP10 displayed the highest CPFPM peak and soil average available P content, but 0.5% HA showed similar effects to 1% and 5% HA proportions in promoting fertilizer P migration and enhancing soil P availability. Considering production costs, technical requirements and yield response, and the results of PLFA, our results suggest that 0.5% HA is the optimal amount for HAP production. Overall, these findings underscore the potential of HAPs as effective P fertilizers, with 0.5% HA serving as a practical and efficient means of enhancing P availability and promoting P migration in soil. However, although the weathered coal HA used in this study was the main raw material for HAP production, the applicability of the findings of this study to HAs of other origins needs to be further investigated.

**Supplementary Materials:** The following supporting information can be downloaded at: https://www.mdpi.com/article/10.3390/agronomy13061576/s1, Table S1: Soil phosphorus activation coefficient in 40–45, 45–50, 50–55 mm soil layer from fertilization layer; Table S2: Soil total PLFA, bacteria, Gram-positive bacteria (Gm+), Gram-negative bacteria (Gm−), fungi, and actinobacteria biomass, fungi: bacteria (F/B), and Gram-positive: Gram-negative (Gm+/Gm−) in the 0–40 mm soil layers around the P application site under NTC, CP, and HAP0.5 treatments; Table S3. Correlation of soil total PLFA, bacteria, Gram-positive bacteria (Gm+), Gram-negative bacteria (Gm−), fungi, and actinobacteria biomass, fungi: bacteria (F/B), and Gram-positive: Gram-negative (Gm+/Gm−) in the 0–40 mm soil layers around the P application site with average TP and AP.

**Author Contributions:** Conceptualization and methodology, B.Z.; software, validation, formal analysis, investigation, J.J., S.Z., Y.Z. and Y.W.; resources, data curation, L.Y., Y.L. and L.Z.; writing—original draft preparation, J.J. and S.Z.; writing—review and editing, B.Z. and X.Y.; visualization, J.J., S.Z., L.Y. and Y.L.; supervision and project administration, X.Y. and Q.X.; funding acquisition, B.Z. and J.J. All authors have read and agreed to the published version of the manuscript.

**Funding:** This research was funded by the National Key Technologies R&D Program of China during the 14th Five-Year Plan period (2023YFD1700205) and the Talent Research Grant Program of Anhui Agricultural University (rc522223).

**Data Availability Statement:** Not applicable.

**Conflicts of Interest:** The authors declare no conflict of interest.

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
