# Peer review of "Effects of Incorporating Different Proportions of Humic Acid into Phosphate Fertilizers on Phosphorus Migration and Transformation in Soil"

_agronomy, doi:10.3390/agronomy13061576_

Round 1

Reviewer 1 Report

The research is very interesting, it is recommended that the authors modify the keywords to have a greater impact and reach on the scientific community.

Author Response

The research is very interesting, it is recommended that the authors modify the keywords to have a greater impact and reach on the scientific community.

Response: Thank you for your affirmation. We have reset the keywords, the new ones are “humic acid-enhanced phosphate fertilizer”, “incorporating proportion”, “humic acid”, “phosphorus migration”, and “phosphorus availability”. Please see p.1, Keywords.

Reviewer 2 Report

The study investigated the effects of incorporating humic acids (HAs) into phosphate fertilizers at different proportions on phosphorus (P) migration and availability in soil. The researchers tested HAs at proportions of 0.5%, 1%, 5%, and 10% (w/w) in the fertilizers.

The results showed that the inclusion of HAs in phosphate fertilizers had significant positive effects on P migration, distance of P movement, and average available P levels in the soil. This suggests that HAs can improve the movement and transformation of P in the soil. The beneficial effects of HAPs (humic acid-phosphate fertilizers) on P migration and availability in soil are attributed, at least in part, to the formation of phosphate monoesters during their production.

Among the various HAP formulations tested, HAP10 (fertilizer with 10% HA) exhibited the highest peak of crop P fertilizer productivity (CPFPM) and the highest average available P content in the soil. However, the study found that even at a proportion as low as 0.5% HA, the effects on promoting fertilizer P migration and enhancing soil P availability were similar to those observed with 1% and 5% HA proportions. Considering factors such as production costs, technical requirements, yield response, and the results of phospholipid fatty acid (PLFA) analysis, the researchers concluded that 0.5% HA is the optimal proportion for HAP production.

In summary, the study highlights the potential of HAPs as effective P fertilizers. Incorporating HAs into phosphate fertilizers, particularly at a proportion of 0.5% HA, can enhance P availability in the soil and promote P migration. This suggests that 0.5% HA could be a practical and efficient approach for improving P availability and movement in agricultural soils.

Comments

Title: I think the word in (Fertilizers) is more suitable

Abstract provides a comprehensive overview and clear indication of the research work conducted.

Introduction is well presented, with minor points to address

Introduction: [27] observed that applying single superphosphate with 45 kg HA ha−1 significantly 65 improved soil water-soluble P contents compared with a single superphosphate application alone

Please mention the quantity of superphosphate used with 45 kg HA ha−1

The same comment for the following: Chen et al. [10] reported that the combination of controlled-release P fertilizers 67 and 45 kg HA ha−1 increased soil AP content by 38.6% compared with controlled-release 68 P fertilizer alone. Du et al. [25] also found that applying monocalcium phosphate fertilizer 69 with 254.8 kg HA ha−1 could increase the distance of P movement by 5 mm. However, 70 Zhao et al. [15]

Material and methods is clear and well organized with some notes

In material and methods part line 181 and 182: the flowing sentence is confusing (The tested soil  was obtained by mixing each soil layer with 5 g of soil, and concentrations of PLFAs were 182 expressed in units of nmol g−1

Results is well presented and inserted figures is clear and informative

In fig 10: the effect of phosphate forms on soil microbial number and diversity should be discussed in more details because according to this results it negatively affect a vital component of soil

Author Response

  1. Title: I think the word in (Fertilizers) is more suitable

Response: Thanks for your advice, we have changed “into Phosphate Fertilizer” with the phrase “into Phosphate Fertilizers” in “Title”, Please see p.1, Title.

  1. Abstract provides a comprehensive overview and clear indication of the research work conducted.

Response: Thank you for your affirmation.

  1. Introduction is well presented, with minor points to address.

Introduction: Shafi et al. [27] observed that applying single superphosphate with 45 kg HA ha1 significantly improved soil water-soluble P contents compared with a single superphosphate application alone. Please mention the quantity of superphosphate used with 45 kg HA ha1. The same comment for the following: Chen et al. [10] reported that the combination of controlled-release P fertilizers and 45 kg HA ha−1 increased soil AP content by 38.6% compared with controlled-release P fertilizer alone. Du et al. [25] also found that applying monocalcium phosphate fertilizer with 254.8 kg HA ha1 could increase the distance of P movement by 5 mm.

Response: We have supplemented the amount of phosphorus applied in the above three literatures with “Shafi et al. [27] observed that applying single superphosphate with 45 kg HA ha−1 significantly improved soil water-soluble P contents compared with a single superphosphate application alone at each increment of P levels from 45 to 112.5 kg P2O5 ha−1. Chen et al. [10] reported that the combination of controlled-release P fertilizers and 45 kg HA ha−1 increased soil AP content by 38.6% compared with controlled-release P fertilizer alone at 75 kg P2O5 ha−1. Du et al. [25] also found that applying monocalcium phosphate fertilizer with 254.8 kg HA ha−1 (at a rate equivalent to 60.9 kg P2O5 ha−1) could increase the distance of P movement by 5 mm.” Please see p.2, Line 67, Lines 69-71.

  1. Material and methods is clear and well organized with some notes.

Response: Thank you for your affirmation.

  1. In material and methods part line 181 and 182: the flowing sentence is confusing (The tested soil was obtained by mixing each soil layer with 5 g of soil, and concentrations of PLFAs were expressed in units of nmol g1.

Response: Sorry for our unclear description. Here, we have re-described the soil sampling method with “Five grams of soil were randomly selected from each of the eight blocks cut from the 0-40 mm soil layer, and then mixed to obtain the tested soil. The concentrations of PLFAs were expressed in units of nmol g−1.” Please see p.6, Lines 199-202.

  1. Results is well presented and inserted figures is clear and informative.

Response: Thank you for your affirmation.

  1. In Fig 10: the effect of phosphate forms on soil microbial number and diversity should be discussed in more details because according to this results it negatively affect a vital component of soil.

Response: Thanks for your advice, we have supplemented the discussion for the results of soil microbial community structure, please see p.14, Lines 429-434, Lines 439-442.

Reviewer 3 Report

Dear Authors,

The article is well written. However, I see some fundamental issues there, due to which it cannot be published in this form.

1) Description of soil and humic acids used - this must be improved. The description of both of these components must be detailed. I am sure that using a different soil or a different source of humic acid (for example, a different molecular weight) would lead to different results. It would help if you also discussed this more in the text.

2) Write clearly in the Conclusion what you came to in your research. Which results are generalizable, and which apply only to your specific research.

Best regards.

Author Response

  1. Description of soil and humic acids used - this must be improved. The description of both of these components must be detailed. I am sure that using a different soil or a different source of humic acid (for example, a different molecular weight) would lead to different results. It would help if you also discussed this more in the text.

Response: I strongly agree with your insight and have added detailed information of the humic acid and soil used in this study. Please see p.2 Lines 93-105; p.3 Lines 115-118; and p.4 Lines 152-153.

  1. Write clearly in the Conclusion what you came to in your research. Which results are generalizable, and which apply only to your specific research.

Response: We have described the possible limitations of our results in the “Conclusion” section. Please see p.14 Lines 458-460.

Round 2

Reviewer 3 Report

Good job. Looks fine now.

Best regards.